# Improved Vulnerability Assessment Table for Retaining Walls and Embankments from a Working-Level Perspective in Korea

**Jaejoon Lee** [1], **Hyunji Lee** [1], **Hongsik Yun** [2], **Chol Kang** [3] **and Moonsoo Song** [1,*]

1. Interdisciplinary Program in Crisis, Disaster and Risk Management, Sungkyunkwan University, Suwon 2066, Korea; lunevocal@naver.com (J.L.); leehyunji94@naver.com (H.L.)
2. School of Civil, Architectural Engineering & Landscape Architecture, Sungkyunkwan University, Suwon 2066, Korea; yoonhs@skku.edu
3. Societal Disaster Response Policy Division, Ministry of the Interior and Safety, Sejong 30128, Korea; aoao0011@gmail.com
* Correspondence: songms0722@gmail.com; Tel.: +82-031-290-7534

**Abstract:** Climate change can lead to unpredictable slope collapse, which causes human casualties. Therefore, Korea has devoted significant effort to the management of slope disasters. The Ministry of the Interior and Safety of Korea, which oversees the safety of the nation's people, has allocated a four-year budget of $557 million to investigate, assess, and maintain steep slope sites. However, there have been fatalities caused by steep slope site evaluations based on inadequate knowledge and a single retaining walls and embankments (RW&E) assessment table. Therefore, the assessment table for RW&E-type steep slopes needs to be improved in terms of its accuracy, simplicity, and ease of use. In this study, domestic and global evaluation methods were reviewed, problems associated with the existing RW&E assessment table were identified, and a focus group interview was conducted. The RW&E assessment table was improved through an indicator feasibility survey and analytic hierarchy processing. The improved assessment table was categorized from one to four classifications to reduce the ambiguity of the evaluation: concrete, reinforced soil-retaining walls, stone embankments, and gabions. This study will provide the sustainability of slope safety and serve as a reference for classification and evaluation criteria across all national institutions that conduct RW&E evaluations.

**Keywords:** disaster management; hazard assessment table; retaining wall; steep slope; analytical hierarchic process (AHP)

## 1. Introduction

Approximately 63% of the landmass of Korea is composed of mountainous regions; the slopes in these regions need to be evaluated during construction processes for efficient land development. Construction activities inevitably require the control of steep slopes, which are managed by various slope management agencies in Korea. In July and August 2020, heavy rain events caused the collapse of the retaining wall at the Changha-eng factory, killing three people and severely injuring one person. These rain events also resulted in the collapse of a reinforced clay retaining wall 10 m away from Chuncheon's residential area. Therefore, assessing physical vulnerability of retaining walls and embankments (RW&E) used on slopes near local roads or residential spaces and designating the areas at risk of collapse is crucial.

The number of steep slopes managed by the Ministry of the Interior and Safety totaled 15,075 as of December 2018, and the numbers presented by each management agency are shown in Table 1. The types of slopes managed by the Ministry of the Interior and Safety include natural slopes, artificial slopes, retaining walls, and embankments. There are currently 14 laws and regulations related to slopes in the Republic of Korea. Representative management agencies include the Ministry of Land, Infrastructure, and Transport, the Ministry of the Interior and Safety, and the Korea Forest Service. As the management

agencies involved vary depending on the length, height, and location of the slopes, the boundaries prescribed by law are divided and managed. Large-scale natural slopes are managed by the Korea Forest Service [1,2], whereas slopes around highways are managed by the Korea Expressway Corporation [3,4] using the assessment table proposed in [5,6]. In addition, the Korea Authority of Land and Infrastructure Safety(KALIS) manages many types of facilities (bridge, rail, road, port, dam, building, tunnel, retaining wall, cut slope) near living areas in accordance with the related regulations in [7,8]. Local roads that may cause traffic and casualties or small slopes next to residential areas that are difficult to evaluate are managed by a central administrative agency called the Ministry of the Interior and Safety.

**Table 1.** Steep slopes and affiliated organizations (Ministry of the Interior and Safety, 2018).

| Affiliated Organization | Number of Steep Slopes |
|---|---|
| Local Government | 10,383 |
| Local Forest Service | 357 |
| Korea Rural Community Corporation | 30 |
| Korea Land & Housing Corporation | 45 |
| Korea Rail Network Authority | 1968 |
| Metropolitan Transit Corporation | 27 |
| Korea National Park Service | 467 |
| Other (e.g., individuals, corporations) | 1798 |
| Total | 15,075 |

As of December 2018, 744 RW&Es were managed by the Ministry of the Interior and Safety of Korea. The locations and statuses of the RW&Es currently under management are shown in Figure 1. The steep slope (natural slopes, artificial slopes, and RW&E) management system comprises the procedures showed in Figure 2. First, steep slopes are selected primarily through resident reports or on-site inspections by working-level officials. Then, repair and reinforcement are carried out after designating a collapse risk zone through an assessment. During this process, assessments made by non-experts using the current system may result in wasted national budgets. Therefore, a suitable assessment table is essential to avoid wasting budgets and to ensure that RW&Es are assessed more accurately. The objective of this study is to create an optimized assessment table capable of evaluating the reliable risk of collapse based on minimal indicators. This table is expected to help non-expert workers conduct on-site evaluations and contribute to policy decisions.

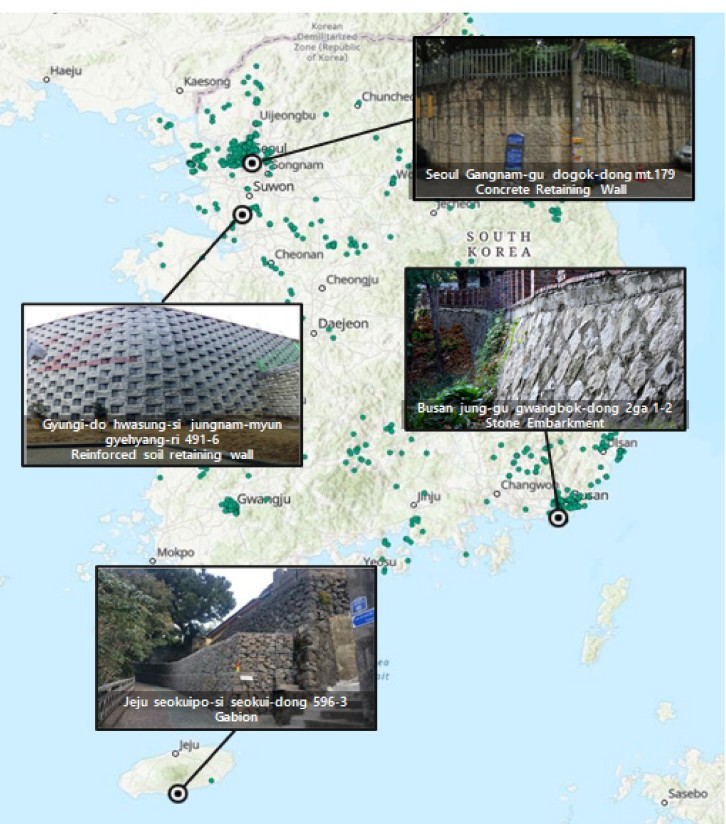

**Figure 1.** Location and type of retaining walls under management in Korea.

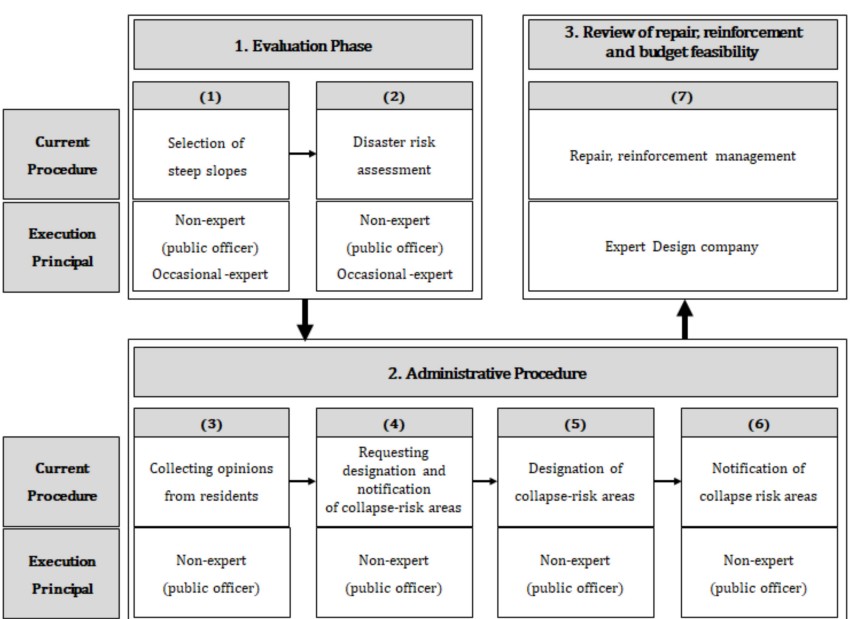

**Figure 2.** Steep slope management system.

## 2. Literature Review

The Korea Authority of Land and Infrastructure Safety (KALIS), an affiliate of the Ministry of Land, Infrastructure, and Transport in Korea, implements professional performance evaluations conducted by experts. Major factors for the retaining wall assessment include subsidence, activity, breakage, damage, drainage, scour, back-filling material, corrosion, and loss, as shown in Tables 2 and 3 [9–11].

**Table 2.** Performance and indicators for retaining walls (Korea Authority of Land and Infrastructure Safety (KALIS), 2019).

| Performance | Sub-Performance | Indicator | Concrete | Reinforced | Stone | Gabion |
|---|---|---|---|---|---|---|
| Safety | Condition safety | Settlement | ○ | ○ | ○ | ○ |
| | | Sliding | ○ | ○ | ○ | ○ |
| | | Drainage, slope angle, rockfall, leaching water | ○ | ○ | ○ | |
| | | Planned linear error (overturing, inclination) | | ○ | ○ | |
| | | Breakage damage cracking | ○ | | | |
| | | Breakage damage (material separation) | ○ | | | |
| | | Cracks | ○ | | | |
| | | Surface degradation (wear, erosion, peeling, fall) | ○ | | ○ | |
| | | Condition of the drain | ○ | | | |
| | | Various phases | ○ | | | |
| | | Rebar exposure | ○ | ○ | ○ | ○ |
| | | Scour | | ○ | ○ | |
| | | Front progressive filling | | | | ○ |
| | | Progressive deformation | | ○ | ○ | |
| | | Washout | | ○ | ○ | |
| | | Spacing | | | ○ | |
| | | Filling concrete condition | | | | ○ |
| | | Filling material loss | | | | ○ |
| | | Wire mesh breakage and damage | | | | ○ |
| | | Wire mesh binding condition | | | | ○ |

Table 2. *Cont.*

| Performance | Sub-Performance | Indicator | | Concrete | Reinforced | Stone | Gabion |
|---|---|---|---|---|---|---|---|
| Safety | Structural safety | External structural safety | Bottom sliding (normal/earthquake) | ○ | ○ | ○ | ○ |
| | | | Circle sliding (normal/earthquake) | ○ | ○ | ○ | ○ |
| | | | Overturing (normal/earthquake) | ○ | ○ | ○ | ○ |
| | | | Settlement | ○ | ○ | ○ | ○ |
| | | | Bearing capacity (normal/earthquake) | ○ | ○ | ○ | ○ |
| | | Internal structural safety | Concrete wall — Design shear strength | ○ | | | |
| | | | Concrete wall — Design bending moment | ○ | | | |
| | | | Reinforced concrete wall — Safety of tensile destruction | | ○ | | |
| | | | Reinforced concrete wall — Safety of breaking | | ○ | | |
| | | | Stone wall — Average width of wall | | | ○ | |
| Durability | Concrete/Reinforced | Deterioration growing | Chloride penetration | ○ | ○ | | |
| | | | Carbonation depth | ○ | ○ | | |
| | | | Concrete quality of surface concrete | ○ | ○ | | |
| | | Deterioration condition | Salty condition | ○ | ○ | | |
| | | | Freezing condition | ○ | | | |
| | Stone condition | | Estimated strength | | | ○ | |
| | | | Weathering degree | | | ○ | |

**Table 3.** Indicators in each country.

| Country/Institution | Potential Indicators from Literature Review |
|---|---|
| South Korea/KALIS | subsidence, activity, breakage, damage, drainage, scour, back-filling material, corrosion, loss |
| Japan/MLIT | drainage facilities, cracks, lateral drifts, subsidence, and fullness |
| Japan/JRA | wall damage (i.e., cracks and defects), fullness, foundation conditions, drainage facilities, structures, auxiliary structures, topographic characteristics, and damage target facilities |
| United States/FHWA | corrosion, deterioration, cracks, damage, subsidence, scour, wire mesh, drainage |
| United States/DOT for each state | capping, draining, joint sidewalls, roadways, slopes, backfills |
| Canada/City of Nanaimo | (wall) tiling, joint, cracks, missing, timber, and staining (soil) settlement, tension cracks, hazard historical data, erosion, and excessive moisture in the backfill |
| France/LCPC | range of influence, drainage facility, structure |

Japan's Ministry of Land, Infrastructure, Transport, and Tourism (MLIT) assesses drainage facilities, cracks, lateral drifts, subsidence, and fullness, whereas the Japan Road Association (JRA) evaluates wall damage (i.e., cracks and defects), fullness, foundation conditions, drainage facilities, structures, auxiliary structures, topographic characteristics, and damage of target facilities.

The U.S. Federal Highway Administration (FHWA) considers corrosion, deterioration, cracks, damage, subsidence, scour, wire mesh, and drainage as evaluation factors. The Wisconsin Department of Transportation (DOT) in the U.S. evaluates retaining walls by dividing them into several categories: capping, draining, joint sidewalls, roadways, slopes, backfills, etc. [12].

The city of Nanaimo in Canada was evaluated by dividing all retaining walls in terms of general and overall designs [13]. The wall types are split into gravity walls, rock-stacked walls, mechanically stabilized earth walls, and reinforced concrete cantilever walls to form a checklist for inspection. A previous study [14] classified earth retaining structures (ERSs) and employed inspections, condition assessments, and ratings to develop a retaining wall inventory and condition assessment system. The field conditions of ERSs were classified as facing, movement, drainage, and exterior. Wall sections were inspected in terms of tiling, joint, cracks, missing, timber, and staining. For soil, settlement, tension cracks, hazard historical data, erosion, and excessive moisture in the backfill were considered as the main factors. Subsequently, the drainage outlets and channels were inspected.

Laboratoire Central des Ponts et Chaussées (LCPC) in France conducted an evaluation by classifying retaining walls based on the range of influence and drainage facility and structure, using indicators similar to those mentioned above. In addition, we reviewed research related to Hong Kong and Colorado, as well as the Minnesota risk assessment table [15–17]. In the literature review, retaining wall management institutions categorize variable retaining wall types, and select and evaluate indicators that fit each type. In order to select indicators through Focus Group Interview (FGI) and proceed with feasibility studies, the potential indicators are extracted regardless of type, through a literature review as shown in Table 3.

Although these indicators were intended to reflect the key elements of the assessment as much as possible, applying them to the Korean management RW&E evaluation system, evaluator, and evaluation results is difficult. Previous studies to improve the suitability of the assessment table for the purpose of the study were referred to during FGI and evaluation indicators in their selection.

Engineering studies evaluating the risks of slopes and studies that increase the reliability of disaster risk assessment tables have been reported. Several studies have evaluated the safety of slopes, structural causes of collapse, and safety of RW&Es using graphical information systems (GIS). With a focus on RW&Es, this study aims to reduce the risk of slope collapse. To achieve this, the evaluation of a prior study on the slope analysis method is deemed necessary.

Previous studies on the assessment of slope collapse risk have employed analyses based on GIS [18–24]. Furthermore, a few studies [25–29] have combined GIS and analytical hierarchic process (AHP), while others [30,31] have employed a combination of statistical and AHP methods. Some studies have also combined GIS with statistical methods [32–34].

The AHP method, which was used to develop the assessment table in this study, is employed in several fields for decision-making. This method, developed by Tomas Saaty in the early 1970s, is a decision-making methodology that allows decision makers to choose the best alternative and determining weights based on their intuitive judgment, thereby enabling complex decisions rationally and efficiently. As AHP is essential for multiple decision-making, several studies have focused on developing this approach. Some studies have also attempted to improve the reliability of the AHP methods used in several fields. A study of random indices that compares theories regarding AHP research has also been reported [35]. Studies are also being conducted to improve the reliability of commonly used AHP methods to determine their importance across various fields. AHP has also been frequently used in studies related to steep slopes, which is the focus of this study. It is mainly used in studies related to sensitivity analyses, vulnerability analyses, relative importance, and grading of slopes. Previous studies [32,36–43] have used the AHP model to assess slope risk and vulnerability.

As shown in the review, the AHP method has been used to derive weights for each slope collapse factor based on spatial data. In terms of disaster risk management, the development of an assessment table for steep slopes based on the selection and extraction of evaluation indicators and the selection of weights has been difficult. This is because agencies in each country have developed and used disaster risk assessment tables through their respective studies.

## 3. Review of Current Assessment Table for RW&Es

In the assessment table for steep slopes in the Republic of Korea, RW&Es are divided into three categories, as shown in Table 4. The first category is divided into sections, namely, collapse risk and social influence. In the second category, collapse risk is divided into three parts: foundation parts, front parts, and outlet. The social influence area is also divided into three parts: circumstances, number of people affected and traffic, and distance from the steep slope land and adjacent facilities [44,45]. Each secondary category was divided into 16 tertiary categories. The results are determined in accordance with Table 5.

Table 4. Retaining walls and embankments (RW&E) assessment table for 29 January 2018. From the Ministry of the Interior and Safety of Korea. (Shaded areas are the improved part in this study.)

| 1st Category | 2nd Category | 3rd Category, Indicators | | 4th Category, Evaluation Standard and Distribution |
|---|---|---|---|---|
| Collapse risk (70) | Foundation part | Subsidence (cm) | | - Selection of indicators and evaluation standard by type through feasibility review Determination of distribution by evaluation standard through AHP |
| | | Lateral Drift (cm) | | |
| | | Scour | Concrete retaining wall | |
| | | | Reinforced soil-retaining wall | |
| | | | Stone embankment | |
| | Front part (major revision object) | Breakage and Damage (mm) | | |
| | | Cracks (mm) | | |
| | | Abrasion/erosion | | |
| | | Exfoliation and Separation of layers (mm) | | |
| | | Rebar exposure (%) | | |
| | | Conduction and Fullness | | |
| | | Efflorescence | | |
| | | Outlet | | |
| Social influence (30) | | Circumstances | | |
| | Number of people affected/road lanes, traffic volume | A 'steep slope' adjacent to the road | Road lane number (one way) | |
| | | | Traffic volume (number of cars/day) | |
| | | Other areas | The estimated number of people affected | |
| | | Distance from 'steep slope' (i.e., to land and adjacent facilities) | | |

**Table 5.** RW&E risk assessment criteria.

| Grade | Risk Assessment Criterion | | | Note |
|---|---|---|---|---|
| | **Natural Slope or Mountainous Region** | **Artificial Slope** | **Retaining Wall & Embankment** | |
| A | 0–20 | 0–20 | 0–20 | • No risk of disaster and less damage in the event of an unexpected collapse |
| B | 21–40 | 21–40 | 21–40 | • No risk of disaster, but it requires periodic management |
| C | 41–60 | 41–60 | 41–60 | • Due to the risk of disaster, continuous inspection must be performed and maintenance plan needs to be established depends on inspector's judgement |
| D | 61–80 | 61–80 | 61–80 | • Need to establish a maintenance plan due to high risk of disasters |
| E | >81 | >81 | >81 | • Due to the high risk of disasters, maintenance plans are required |

The problem is that the current assessment table, which is often presented in different forms, is evaluated by a single assessment table regardless of retaining wall types, and some of the assessment items are inappropriate for assessing disaster risk. First, for items such as 'Conduction and Fullness' and 'Efflorescence', the scores are always five points, regardless of the type of retaining wall, as shown in Table 6. Second, the items in 'Cracks' differ in their scores depending on the size of the crack, as shown in Table 7. However, in the case of retaining walls, the patterns of cracks vary in each type, such as concrete retaining walls, reinforced soil-retaining walls, stone embankments, and gabion retaining walls. These problems with the assessment table have caused considerable confusion among working-level officials, making accurate and effective assessments difficult.

**Table 6.** Items of 'Conduction ● Fullness' and 'Efflorescence' in the assessment table for RW&Es.

| Collapse risk (70) | Front part | Conduction ● Fullness | Non-existence 0 | Existence 5 |
|---|---|---|---|---|
| | | Efflorescence | Non-existence 0 | Existence 5 |

**Table 7.** Items of 'Crack' in the assessment table for RW&Es.

| Collapse risk (70) | Front part | Cracks (mm) | <0–0.1 | >0.1–<0.2 | >0.2–<0.3 | >0.3–<0.5 | >0.5 |
|---|---|---|---|---|---|---|---|
| | | | 1 | 3 | 5 | 7 | 10 |

The Republic of Korea has revised its assessment table for RW&Es seven times over the past 11 years; however, only a few of the evaluation indicators and scores have been revised as shown in Table 8 [46]. It is our contention that further revision of the assessment table is required.

This study primarily focuses on the derivation of evaluation indicators for the front part, and the allocation of points is based on the type of retaining wall. Steep slopes threaten the safety of people, and their maintenance is expensive. The Ministry of Public Administration and Security spent $75 million in 2019 alone for the repair of such slopes. Therefore, developing a highly accurate and reliable assessment table is crucial.

**Table 8.** Risk assessment table for RW&Es (●: Whether to include evaluation indicator, △: Change evaluation criteria, ■: Change evaluation points), (1) Starting with 21 April 2017, the items under scour will be further subdivided into three categories: concrete retaining walls, reinforced soil-retaining walls, and stone embankments. (2) From 3 April 2020, cracks are further subdivided into three categories: concrete retaining walls, reinforced soil-retaining walls, and stone embankments. (3) The exfoliation items are integrated into exfoliation and separation of layers. (4) From 3 April 2020, conductive/fullness items are subdivided into three categories: concrete retaining walls, reinforced soil-retaining walls, and stone embankments. (5) Rename it to 'The estimated number of people affected'. (6) Rename it to 'Distance from building'. (7) Rename it to 'Distance from 'steep slope'-land and the adjacent facilities' -Starting from 29 January 2018, investigator correction score item 4 will be newly established.

| Division | | | 09.09 | 11.11.25 | 15.4.2 | 15.10.20 | 17.4.21 | 17.7.26 | 18.1.29 | 20.4.3 |
|---|---|---|---|---|---|---|---|---|---|---|
| Collapse risk (70) | Foundation part | Subsidence | ● | ● | ● | ● | ● | ● | ● | ● |
| | | Lateral Drift | ● | ● | ● | ● | ● | ● | ● | ● |
| | | Scour | ● | ● | ● | ● | ●(1) | ● | ● | ● |
| | Front part | Breakage and Damage | ● | ● | ● | ● | ●△ | ● | ● | ● |
| | | Cracks | ● | ● | ● | ● | ●△ | ● | ● | ●(2) |
| | | Abrasion/erosion | ● | ● | ● | ●■ | ● | ● | ● | ● |
| | | Exfoliation | ● | ● | ● | | | | | |
| | | Exfoliation and Separation of Layers | ● | ● | ● | ●(3) | ● | ● | ● | ● |
| | | Rebar exposure | ● | ● | ● | ● | ● | ● | ● | ● |
| | | Efflorescence | | ● | ● | ●■ | ● | ● | ● | ●△■ |
| | | Conduction and Fullness | | | | ● | ● | ● | ● | ●(4) |
| | | Chloride | ● | | | | | | | |
| | | Outlet | ● | ● | ● | ●■ | ● | ● | ● | ● |
| Social influence (30) | | Circumstances | ● | ● | ● | ● | ● | ● | ● | ● |
| | | The estimated number of people affected | ● | ●(5) | ● | ● | ● | ● | ● | ● |
| | | Distance from 'steep slope'-land and the adjacent facilities | ● | ● | ●△■ | ●△■(6) | ●(7) | ● | ● | ● |
| | | Traffic volume | | | ● | ● | ● | ● | ● | ● |
| | | Road lane number | | ● | ●■ | ● | ● | ● | ● | ● |

## 4. Development of the Assessment Table

In this study, a meeting with the managers of three local government bodies and a survey involving 21 experts were conducted. Selection of the RW&E evaluation indicators and criteria were conducted based on an FGI with experts, and the opinions of the participants were gathered simultaneously. A total of 21 experts participated in the AHP to determine the scores for the evaluation criteria. The schematic of this research is shown in Figure 3.

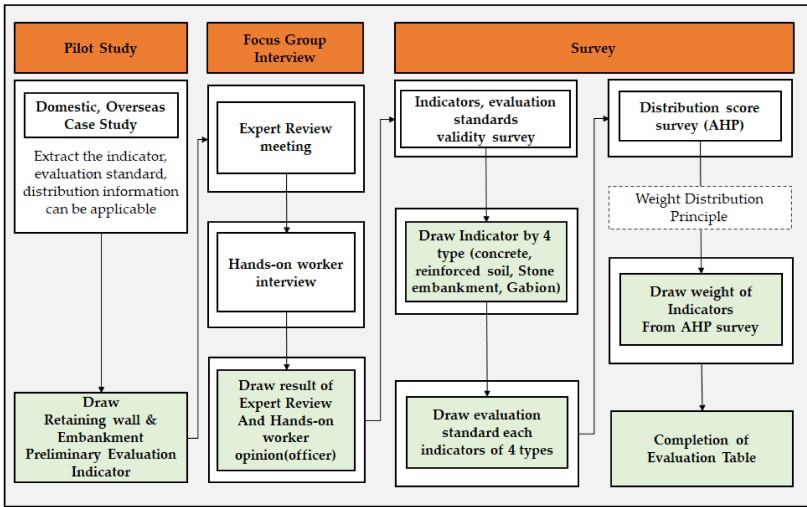

**Figure 3.** Schematic of this study.

## 5. Summary of Approaches for the Development of the Assessment Table

The evaluation indicators were largely divided into collapse risk and social influence indicators, and the collapse risk indicators were further divided based on the foundation, front, and outlet parts. The need for a review of the front part of the collapse risk portion of the RW&E disaster risk assessment table will be raised, and the evaluation indicators and criteria for the front part will be classified depending on the type of retaining wall. In developing the evaluation indicators, criteria, and distribution, we aimed to satisfy field adaptability, usability for hands-on workers, and support for policy decision-making. The assessment table improvement direction is described in Figure 4.

### 5.1. Deriving Evaluation Indicators and Standards

Based on an analysis of Korean and global RW&E evaluation methods, we drafted three categories and three layers, as shown in Table 9. If the evaluation table is completely changed, hands-on workers might not be able to understand it, and appropriate evaluations may not be possible. Therefore, considerable efforts were made during the FGI to produce a draft of the preliminary evaluation indicators. The draft was created through interviews with experts and hands-on workers.

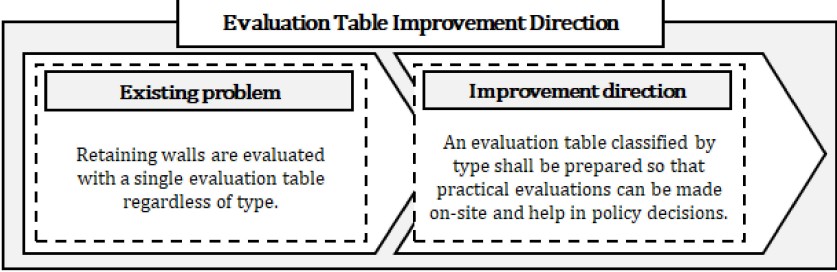

**Figure 4.** Assessment table improvement direction.

**Table 9.** Results of layers 1, 2, and 3 reflecting the expert feasibility review. (The shaded area is excluded because it failed the feasibility study.)

| Divisions | Indicators | Evaluation Standards | | | | |
|---|---|---|---|---|---|---|
| **1st Layer** | **2nd Layer** | **3rd Layer** | | | | |
| Concrete | Breakage and Damage (mm) | None | >0–<5 | >5–<10 | >10–<20 | >20 |
| | Cracks (mm) | <0–0.1 | >0.1–<0.2 | >0.2–<0.3 | >0.3–<0.5 | >0.5 |
| | Abrasion/erosion | None | Slight | Slightly bad | Bad | Very bad |
| | Exfoliation and Separation of layers (mm) | 0–10 | 11–15 | 16–20 | 21–25 | >26 |
| | Rebar exposure(%) | 0 | 0.1–1 | 1.1–3 | 3.1–5 | >5.1 |
| | Efflorescence | None–partial discovery | Found many places–Distributed carefully and widely | | | |
| Reinforced soil-retaining wall | Breakage, Damage, and Cracks | None | Surface damage | Surface damage, Damage progressible status | Partial damage and damage scale expansion status | Very Severe, Broken Function |
| | Loss | Non-existent | Observed | | | |
| | Separation | None | Slight | Slightly bad | Bad | Very bad |
| | Conduction ● Fullness | None | Slight Inactive state | Slightly bad State in progress | Bad Condition affecting structural stability | Very bad Condition that significantly affects structural stability |

**Table 9.** *Cont.*

| Divisions | Indicators | Evaluation Standards | | | | |
|---|---|---|---|---|---|---|
| 1st Layer | 2nd Layer | 3rd Layer | | | | |
| Stone embankment | Breakage, Damage, and Cracks | None | Surface damage | Surface damage, Damage progressible status | Partial damage and damage scale expansion status | Very Severe, Broken Function |
| | Loss | None | Slight | Slightly bad | Bad | Very bad |
| | Separation | None | Slight | Slightly bad | Bad | Very bad |
| | Conduction • Fullness | None | Slight Inactive state | Slightly bad State in progress | Bad Condition affecting structural stability | Very bad Condition that significantly affects structural stability |
| | Filling concrete | Good | Micro -crack generation | Partial crack occurrence not serious condition | Filled concrete, weathering condition | Lost state |
| | Delete (hard to determine grade) | | | | | |
| Gabion | Loss of filling material | None | >0–<5 | >5–<10 | >10–<20 | >20 |
| | Progressive deformation | None | Slight | Slightly bad | Bad | Very bad |
| | United wire mesh condition | Strain-proof wire mesh with three or more layers horizontally and vertically | Strain-proof wire mesh with two or more layers horizontally and vertically | Strain-proof wire mesh with one or more layers horizontally and vertically | Strain-proof wire mesh with one horizontally and vertically | No strain-proof wire mesh |
| | Fullness | None | Slight Inactive state | Slightly bad State in progress | Bad Condition affecting structural stability | Very bad Condition that significantly affects structural stability |
| | Wire Breakage | None | Slightness | Slightness Possible additional damage | Breakage progress Loss of fillings in progress | Wire breakage Impaction on structural stability |

The evaluation table used in the past was divided into three categories (concrete, reinforced soil retaining wall, and stone embankment). Based on the previous evaluation table, the FGI interview was conducted. As a result, four categories (concrete, reinforced soil retaining wall, and stone embankment, 'gabion') were selected. Gabion was added because it occupies a large proportion of the slope protection facilities managed by the Ministry of Interior and Safety. In addition, a feasibility study was conducted for each category.

The 'Front part' in Table 4, which was previously evaluated as one type, was divided into four types: concrete retaining wall, reinforced soil-retaining wall, stone embankment, and gabion in Table 9. For the second layer, six types of concrete, four types of reinforced soil, six types of stone embankment, and five types of gabion were selected, and feasibility studies were conducted for the third layer. Table 9 shows the results of feasibility review. There were six participants in the FGI for managing slopes from each institution, and 21 participants in the feasibility studies. The RW&E experts included a Ph.D. from the Korea Forest Service, the Korea Institute of Civil Engineering and Building Technology, the Korea Expressway Corporation, and the Ministry of the Interior and Safety; all of which manage slopes in Korea.

The feasibility study comprised a 5-point scale of first, second, and third layers to evaluate suitability, and the average of the evaluation scores was calculated to determine whether items with an average score of 3.5 or higher were suitable. In addition, open surveys were added to accommodate the diverse opinions of experts to ensure that important opinions could be presented when making decisions during feasibility studies. According to the survey, securing the feasibility of one indicator and the feasibility of seven evaluation standards was difficult. One indicator in the gabion section was the weathering degree of the rock. There was an opinion that the deterioration and adhesion state of the cement, the bonding material, dominates the stability. Therefore, it reflected the result that it should be excluded as not having a significant impact on potential collapse. The evaluation criteria were then restated by referring to open-type advice on the uncertainty of the seven evaluation standards. The results are shown in Table 9. The use of difficult-to-judge terms was a major problem. No accurate standard exists for qualitative judgment criteria (e.g., none, minor, severe, and very serious, among others). It is believed that this problem will be solved by providing evaluation guidelines.

### 5.2. Weight Analysis by Indicators

### 5.2.1. Weight Analysis Methodology

Twenty-one experts responded to the survey for AHP. A reliable measure of assessment is required for successful weight selection. Therefore, this study used the nine-point scale of AHP proposed by Saaty [47,48]. The consistency index (CI) is an indicator that shows how consistently the performer records the results, indicating that the CI verifies the logical inconsistency of the response and that the closer it is to zero, the more consistent it is. As the number of attributes increases, the CI value increases; therefore, verification procedures are performed based on the consistency ratio (CR) value. The CR identified the CI divided into the random consistency index (RI) in [35]. Reference [49] states that a CR < 0.1 is relatively consistent in thinking. Reference [50] states that a consistency assessment should be assessed using CR values, and an interviewed person with a CR value less than 0.1 should be adopted to determine the relative weight and overall CR values as the arithmetic mean.

### 5.2.2. Results of Weight Analysis

- Concrete Retaining Wall

The analysis results showed that 17 of the 21 CRs, excluding 4 with a CR value of 0.1 or higher, showed high consistency with a total CI value of 0.005 and a CR value of 0.004. The weight of breakage and damage is 0.231, the crack is 0.224, the abrasion and erosion is 0.095, the exfoliation and separation of layers is 0.132, the rebar exposure is 0.244, and the

efflorescence is 0.073. The most important indicator of concrete retaining wall is the rebar exposure.

- Reinforced Soil-Retaining Wall

According to the analysis, 21 experts conducted a consistent survey, with a CI value of 0.0015 and a CR value of 0.0017. The weight of the breakage, damage and crack is 0.196, the loss of back filling material is 0.210, the separation is 0.150, and the conduction and fullness is 0.444. The most important indicator of the reinforced soil retaining wall is the conduction and fullness.

- Stone Embankment

In the case of the stone embankment, the consistency of 16 expert surveys was confirmed and the results were analyzed. The total CI value of the 16 experts was 0.0008, and the CR value was 0.00071, indicating high consistency in the survey. The weight of the breakage, damage, and crack is 0.195, the loss is 0.195, the separation is 0.128, the conduction and fullness is 0.338, and the filling concrete condition is 0.143. The most important indicator of stone embankment is also conduction and fullness.

- Gabion

In the gabion AHP survey, 18 experts exhibited consistency that was used in the results. The CI value of the overall result was 0.0056, and the CR value was 0.005. The weight of loss of filling material is 0.157, the wire breakage is 0.186, the progressive deformation is 0.202, the united wire mesh condition is 0.224, and the fullness is 0.231. The most important indicator of gabion is the united wire mesh condition

Figure 5 shows the schematic and results of weight analysis. The individual values and Table 10 were used to verify the reliability through the AHP survey.

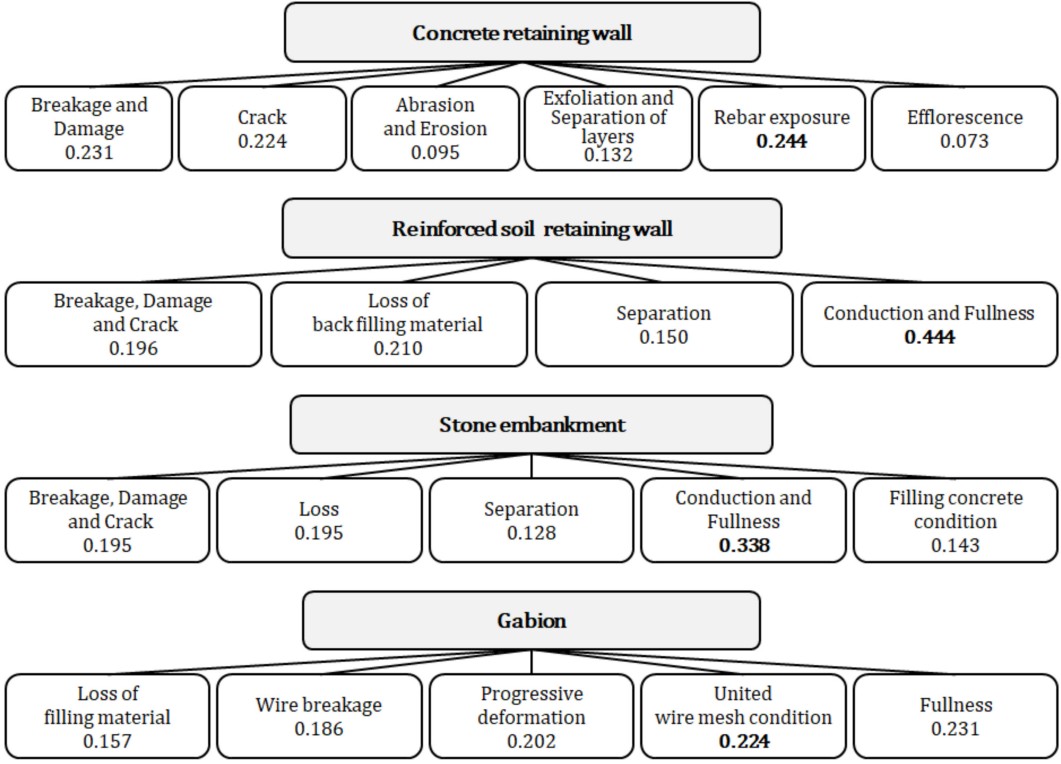

**Figure 5.** Schematic and weights of Analytic Hierarchy Process (AHP) evaluation indicators.

**Table 10.** CR (CI/RI) for verifying the reliability (CR: consistency ratio, CI: consistency index, RI: random consistency index) (The shaded areas have a CR value of 0.1 or more.)

| Expert's No. | Concrete | Reinforced Soil | Stone Embankment | Gabion |
|---|---|---|---|---|
| RI | 1.21 | 0.90 | 1.12 | 1.12 |
| 1 | 0.028234 | 0.017057 | 0.027232 | 0.015268 |
| 2 | 0.091508 | 0.096377 | 0.028571 | 0.047411 |
| 3 | 0.053532 | 0.000000 | 0.027679 | 0 |
| 4 | 0.029911 | 0.044327 | 0.031339 | 0.042054 |
| 5 | 0.112556 | 0.053369 | 0.03402 | 0.056518 |
| 6 | 0.013113 | 0.007670 | 0.007411 | 0.008125 |
| 7 | 0.013113 | 0.011497 | 0.015268 | 0.015268 |
| 8 | 0.035153 | 0.068682 | 0.053839 | 0.016786 |
| 9 | 0.019169 | 0.032781 | 0.040089 | 0.005893 |
| 10 | 0.033250 | 0.026258 | 0.120893 | 0.063571 |
| 11 | 0.296516 | 0.000000 | 0.002946 | 0.022946 |
| 12 | 0.077371 | 0.007634 | 0.033929 | 0.017589 |
| 13 | 0.018169 | 0.003844 | 0 | 0 |
| 14 | 0.316710 | 0.077729 | 0.161161 | 0.091518 |
| 15 | 0.068024 | 0.097966 | 0.121071 | 0.003723 |
| 16 | 0.071177 | 0.059134 | 0.071786 | 0.194554 |
| 17 | 0.009613 | 0.007643 | 0.004464 | 0.002946 |
| 18 | 0.000000 | 0.072272 | 0.004464 | 0 |
| 19 | 0.021241 | 0.045084 | 0.043661 | 0.06625 |
| 20 | 0.192430 | 0.193428 | 0.189107 | 0.189107 |
| 21 | 0.098955 | 0.044128 | 0.105893 | 0.141071 |
| <0.1 | 0.004 | 0.0017 | 0.00071 | 0.005 |

*5.3. Producing Weights by Assessment Table Indicators*

5.3.1. Principle of Scoring Criteria by Indicators

In the AHP analysis, the weights for each indicator were distributed based on 45 points, similar to the existing assessment table. If there was no beginning for each indicator, 0 points were processed. If the beginning of the evaluation indicator was expressed numerically rather than on the basis of 'None,' the weighted total points were evenly distributed. Consistency verification of the personal survey results was performed by checking the CR values within the 0.1 range.

5.3.2. Results

The results of the assessment table in the 'Front part' for RW&Es are shown in Table 11. The order of importance in concrete is rebar exposure, breakage, and cracking, among others. In reinforced soil-retaining walls, soil material fullness is the most important factor, and for stone embankments, fullness is the most important factor. These are the most important factors because the loss of the fillings degrades the stability of the entire structure. In the gabion division, wire breakage (11.2 p), fullness (9.6 p), and progressive deformation (9.8 p) were not significantly different. However, the loss of the filling material score was low, unlike other retaining wall types.

**Table 11.** Improvement of RW&Es in the disaster risk assessment table of steep slopes (proposed).

| Indicator by Type | | Risk of Collapse (45) | | | Front Part | |
|---|---|---|---|---|---|---|
| Division | | Evaluation Standard and Distribution | | | | |
| Concrete | Breakage and Damage (mm) | None | >0–<5 | >5–<10 | >10–<20 | >20 |
| | | 0 | 2.65 | 5.3 | 7.95 | 10.5 |
| | Cracking (mm) | <0–0.1 | >0.1–<0.2 | >0.2–<0.3 | >0.3–<0.5 | >0.5 |
| | | 1.92 | 3.84 | 5.76 | 7.68 | 9.5 |
| | Abrasion/erosion | None | Slight | Slightly bad | Bad | Very bad |
| | | 0 | 1.05 | 2.1 | 3.15 | 4.2 |
| | Exfoliation and Separation of layers (mm) | 0–10 | 11–15 | 16–20 | 21–25 | >26 |
| | | 1.16 | 2.32 | 3.48 | 4.64 | 5.8 |
| | Rebar exposure (%) | 0 | 0.1–1 | 1.1–3 | 3.1–5 | >5.1 |
| | | 0 | 2.9 | 5.8 | 8.7 | 11.6 |
| | Efflorescence | None ~partial discovery | Found many places ~Distributed carefully and widely | | | |
| | | 0 | 3.4 | | | |
| Reinforced soil -retaining wall | Breakage, Damage (mm), and Cracks | None | Surface damage | Surface damage, Damage progressible status | Partial damage and damage scale expansion status | Very Severe, Broken Function |
| | | 0 | 2.125 | 4.25 | 6.375 | 8.5 |
| | Loss | Non-existent | Existence | | | |
| | | 0 | 9.6 | | | |
| | Separation | None | Slight | Slightly bad | Bad | Very bad |
| | | 0 | 1.725 | 3.45 | 5.175 | 6.9 |
| | Fullness | None | Slight Inactive state | Slightly bad State in progress | Bad Condition affecting structural stability | Very bad Condition that significantly affects structural stability |
| | | 0 | 5 | 10 | 15 | 20 |

**Table 11.** *Cont.*

| Indicator by Type | | Risk of Collapse (45) | | | Front Part | |
|---|---|---|---|---|---|---|
| **Division** | | **Evaluation Standard and Distribution** | | | | |
| Stone embankment | Breakage, Damage (mm), and Cracks | None | Surface damage | Surface damage, Damage progressible status | Partial damage and damage scale expansion status | Very severe, Broken Function |
| | | 0 | 2.2 | 4.4 | 6.6 | 8.8 |
| | Loss | None | Slight | Slightly bad | Bad | Very bad |
| | | 0 | 2.2 | 4.4 | 6.6 | 8.8 |
| | Separation | None | Slight | Slightly bad | Bad | Very bad |
| | | 0 | 1.45 | 2.9 | 4.35 | 5.8 |
| | Fullness | None | Slight Inactive state | Slightly bad State in progress | Bad Condition affecting structural stability | Very bad Condition that significantly affects structural stability |
| | | 0 | 3.8 | 7.6 | 11.4 | 15.2 |
| | Filling concrete | Good | Micro -crack generation | Partial crack occurrence Not serious condition | Filled concrete Weathering condition | Lost state |
| | | 0 | 1.6 | 3.2 | 4.8 | 6.4 |
| | Weathering degree of rock | Delete | | | | |

**Table 11.** *Cont.*

| Indicator by Type | | Risk of Collapse (45) | | Front Part | | |
|---|---|---|---|---|---|---|
| **Division** | | **Evaluation Standard and Distribution** | | | | |
| Gabion | Loss of filling material | None | >0–<5 | >5–>10 | >10–<20 | >20 |
| | | 0 | 1.625 | 3.25 | 4.875 | 6.5 |
| | Progressive deformation | None | Slight | Slightly bad | Bad | Very bad |
| | | 0 | 2.225 | 4.45 | 6.675 | 8.9 |
| | United wire mesh condition | Strain-proof wire mesh with three or more layers horizontally and vertically | Strain-proof wire mesh with two or more layers horizontally and vertically | Strain-proof wire mesh with one or more layers horizontally and vertically | Strain-proof wire mesh with one horizontally and vertically | No strain-proof wire mesh |
| | | 1.76 | 3.52 | 5.28 | 7.04 | 8.8 |
| | Fullness | None | Slight Inactive state | Slightly bad State in progress | Bad Condition affecting structural stability | Very bad Condition that significantly affects structural stability |
| | | 0 | 2.4 | 4.8 | 7.2 | 9.6 |
| | Wire Breakage | None | Slight | Slight Possible Additional Damage | Breakage progress Loss of fillings in progress | Wire breakage Impaction on structural stability |
| | | 0 | 2.8 | 5.6 | 8.4 | 11.2 |

- Concrete Retaining Wall

The results showed the breakage and damage score of 10.5, the crack score of 9.5, the abrasion and erosion score of 4.2, the exfoliation and separation of layers score of 5.8, the rebar exposure score of 11.6, and the efflorescence score of 3.4.

- Reinforced Soil-Retaining Wall

It was confirmed that conduction fullness had the greatest impact on the safety of reinforced soil-retaining walls, with scores of 8.5 for the breakage, damage, and crack, 9.6 for the loss of backfilling material, 6.9 for the separation, and 20 for the conduction and fullness.

- Stone Embankment

According to the analysis, the scores for each indicator were: the breakage, damage, and cracks was 8.8, the loss was 8.8, the separation was 5.8, the conduction and fullness was 15.2, and the filling concrete was 6.4.

- Gabion

According to the analysis, the scores for each indicator were: the loss of filling material of 6.5, the wire breakage of 11.2, the progressive deformation of 8.9, the united wire mesh condition of 8.8, and the fullness of 9.6. It was confirmed that fullness and wire breakage are important indicators regarding the risk of collapse of gabion structures.

## 6. Discussion and Conclusions

South Korea, which has a small area, often needs to level slopes to create roads and residences. Therefore, several retaining wall structures have been created. RW&Es, which are types of steep slopes, are managed by the Ministry of the Interior and Safety, which focuses on local roads and residential areas. Due to recent climate change, rainfall intensity and volume have greatly increased, necessitating accurate safety assessments of aging facilities.

This study improved the assessment indicators, and the following conclusions were drawn:

- The RW&E hazard assessment table for the front part, which was previously evaluated as a single type, was classified into four types.
- The person in charge of the assessment should select an evaluation indicator to ensure accuracy with minimal indicators and develop an evaluation table.
- The weights of each evaluation indicator were derived from an expert AHP analysis. As a result, objectivity and scientificity were guaranteed, and existing evaluation indicators were improved.

If the evaluation of RW&Es is required to support repair and reinforcement efforts and policy decision-making, it is expected that the improvement of the RW&E assessment table will be made more efficiently based on the developed assessment table as a result of this study. The assessment tables used by the international community helped identify the difficulties of using professional terminology, the ambiguity in determining the situation, and the need for experts during the assessment. Korea's steep slopes are included in a management system that selects and evaluates these slopes based on residents' reports or the observations of working-level officials. Under the current system, wherever experts are not present to select and evaluate sites, complex assessment tables are used, which may diminish the accuracy of the evaluation. Considering that all types of RW&Es are complex, risk assessment studies should be based on accurate classifications as much as possible.

It is also necessary to achieve the purpose of protecting vulnerabilities, such as the surrounding population, buildings, and roads, by improving social influence as well as the portion of collapse risk improved in this study. The indicators in the assessment table alone, which are the results of the study, are difficult to evaluate. A solution will be presented to exclude subjective judgments when conducting the assessment. We also plan to reduce

ambiguity by presenting evaluation guidelines, similar to a previous study [51], which can be used in practice. The use of the improved assessment table for determining the collapse risk of RW&Es is expected to contribute to accurate evaluations and reduced expenses for the refurbishment of steep slopes, which can amount to $600 billion over a period of four years. We also expect that the assessment tables developed in this study will serve as a reference for retaining walls in foreign countries where experts cannot directly evaluate such structures to the benefit of hands-on workers.

This study is derived from a research and development project conducted from 2019 to 2022 with funding from the Ministry of Public Administration and Security. In 2019 and 2020, a revised study of the evaluation table for retaining wall was conducted. In 2021, we will select a target research area and directly apply the evaluation table of results of this study. Minor corrections can be made through field application. Detailed 4 guidelines can be revised through on-site verification, but the results of indicators and points will not change.

**Author Contributions:** Conceptualization, J.L. and M.S.; Methodology, J.L.; FGI Interview, H.Y. and J.L.; field hands-on worker interview, J.L. and H.L.; Indicator Survey, M.S. and H.L.; AHP Survey, J.L. and H.L.; Project Administration, C.K. and H.Y.; Data curation, C.K. and H.L.; Result data acquisition, M.S. and H.L.; writing—original draft preparation, J.L.; visualization, H.L.; funding acquisition, J.L. and H.Y.; supervision, M.S. and H.Y. All authors have read and agreed to the published version of the manuscript.

**Funding:** This research was supported by a grant (2019-MOIS33-005) of Lower-level and Core Disaster-Safety Technology Development Program funded by the Ministry of Interior and Safety (MOIS, Korea).

**Institutional Review Board Statement:** Not applicable.

**Informed Consent Statement:** Not applicable.

**Data Availability Statement:** No new data were created or analyzed in this study. Data sharing is not applicable to this article.

**Conflicts of Interest:** The authors declare no conflict of interest.

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
