# Peer review of "Improved Vulnerability Assessment Table for Retaining Walls and Embankments from a Working-Level Perspective in Korea"

_sustainability, doi:10.3390/su13031088_

Round 1

Reviewer 1 Report

The paper presents the development of a method for vulnerability assessment of retaining walls and embankments. The paper has the potential to be very useful to the relevant industry. The developed assessment table seems to have enough details but yet practical enough to allow the assessment to be carried out.

However, I must say that in order for this paper to be of a publishable standard, a significant improvement needs to be made on the manuscript. Sections 5.2 and 5.3,  which I believe are the most important sections, are currently not well written. I do not know how the authors came up with the values and how they determine which indicator is most important. There has to be a clear definition of CI, IR, CR and how they can be determined. What are these divisions that went from 1 to 21, and the numbers in Table 10. As they are used (I believe) to determine which indicators are important. It is very important to be very clear how the authors arrive at these figures.

Further, there are a few issues with the way the sentences were constructed, which make them difficult to understand. For example, lines 63-67, lines 114 - 115 ( how do the collapse hazard factors have the same collapse mechanism - structures may exhibit a similar collapse mechanism but not factors), lines 122 - 134 (the paragraph has to be arranged - starting with explaining what AHP is, what does the method involve, where it has been used and how and what is the limitation). Other similar issues with phrasing: line 37 "evaluated" is a more appropriate verb than "addressed", line 52 "human damage" is not an appropriate term, line 69 "exact" is not an appropriate word (perhaps reliable is more appropriate here), line 112 " were referred to as FGI" should be "were referred to during FGI", Figure 3 "oversea" should be "overseas". Please note that the list is not comprehensive. The authors need to improve phrasings used in the manuscript and have the paper proofread. 

Other technical and editorial queries are as follows:

Title - I think the word Hazard is not appropriate. Hazard is about what causes harm - in this paper, might be heavy rain, site conditions, which are not covered by the assessment. Vulnerability assessment, I believe, is more appropriate as the authors are in fact assessing the conditions of the structure and determine its vulnerability.

line 81 - KISTEC is the best institution .... - this statement should be substantiated.

Table 3  should summarise all published literature mentioned in lines 87-103. Also, it can use another column on different types of structures covered by the assessments.

Tables 4 and 5 should be placed closer to Section 3 where they were first cited. 

Section 3, paragraphs 2 and 4 - I am not clear on what the authors meant with <1>, <2>.

Section 5.1, line 195 - what did the authors meant with feasibility studies here. If the authors conducted feasibility studies, this needs to be discussed further in terms of what has actually been done, the method, the purpose, the outcomes.

Line 199 - the authors should articulate who the participants of the focus group and justify why they were chosen. For example, why were the relevant industries not included?

Table 10 and 11 should be swapped as the authors referred to Table 11 first in the text.

Reviewer 2 Report

The study proposes an improved assessment evaluation table for Retaining walls and Embankments where the risk of collapse is divided in 4 categories of RW&Es to reduce the ambiguity of the evaluation and an indicator feasibility survey and AHP was conducted.

The work is interesting and acceptable for publication considering the relevance at national scale. However, it needs major revision.

Particularly the manuscript needs improvement in the structure and the explanation of some parts described in the procedure. Further efforts need to be done to link the text to the Tables proposed (some of them are not even mentioned in the text) to help the reader to easily follow the various paragraphs of the paper.

Line-by-line comments to the Authors are below:

Line 39-40: Consider to change from "caused the retaining wall at the …… to collapse" to " caused the collapse of the retaining wall at the…"

Line 41: It is not clear if it is one single rain event or heavy rains (as written earlier) that could also mean many rain events. Please clarify.

Line 42: The Authors say: "Assessing retaining walls…". Please specify the assessing of what of retaining walls? (i.e. assessing the suitability of retaining walls? the stability of retaining walls? the risk of collapse of retaining walls?).

Line 51: The Authors mention six types and list 5 of them in brackets (i.e. road, rail, port, dam, structure, etc.). Please consider adding also the 6th, since already 5 types are listed.

Line 54-56: It is not clear the role of the Ministry of the Interior and Safety, are all the representative management agencies falling under this Ministry? Please consider to move this sentence starting with "the number of steep slopes…" as introduction of this paragraph, for example at line 44.

Line 60: please check the font size of "Steep slope" and "management system"

Line 61: please change "following Figure 2" in "procedures showed in Figure 2"

Line 64: Are the non-specialists the same of non-expert in Figure 2? If yes, please consider to use only one term.

Line 81: Change "that specializes" in " specialized"

Table 3: First of all Table 3 should be cited somewhere in paragraph 2 Literature Review.

Line 90: add DOT in brackets after "Department of Transportation"

Line 179: It is not clear why it is used the future "will be raised". Was this a proposal for a change of the assessment table that hasn't been accepted yet? Please clarify.

Line 188: What the Authors say is not shown in Table 8, only in the captions of Table 8 it is explained how the categories were changed from the 2020's revision. Please delete the reference to Table 8.

Line 192: please clarify that you are referring to Layer 1 of Table 9.

Line 193: on the base of what these 4 categories were introduced? Was there a list of categories presented from which the surveyed people should select the top 4? Please the Authors should clarify.

Line 194-196: how were the layers 2 selected for each category? Please clarify

Line 221: If the number are the same as the previous paragraph, please here specify that there were 21 PhDs (the experts) and 6 public officers (responsible of slope management). As it is now it seems that there were 21 phDs and who knows how many public officers.

Line 250: the experts are always the same as before. No need to specify at each paragraph. It seems here that the survey was sent out to an infinite number of experts and 21 replied back.

Table 10: please explain in the captions what is R.I. for in the first line.

Round 2

Reviewer 2 Report

The authors have properly addressed the comments, improving the overall quality of the manuscript.

I recommend the Authors to add in the introduction or in the paragraph of discussions the answer to question n.13 provided in the cover letter, after a minor adjustment in the content.

"This study is a study derived from a research and development project conducted from 2019 to 2022 with funding from the Ministry of Public Administration and Security. In 2019 and 2020, a revised study of the evaluation table for retaining wall was conducted. In 2021, we will select a target research area and directly apply the evaluation table of results of this study. Minor corrections can be made through field application. Detailed
4 guidelines can be revised through on-site verification, but the results of indicators and points will not change."
